# Method Validation for Determination of Thallium by Inductively Coupled Plasma Mass Spectrometry and Monitoring of Various Foods in South Korea

**DOI:** 10.3390/molecules26216729

**Published:** 2021-11-06

**Authors:** Yeon-hee Kim, Wook-jin Ra, Solyi Cho, Shinai Choi, Bokyung Soh, Yongsung Joo, Kwang-Won Lee

**Affiliations:** 1Department of Biotechnology, College of Life Sciences and Biotechnology, Korea University, Seoul 02841, Korea; helly27@korea.ac.kr (Y.-h.K.); ra228@korea.ac.kr (W.-j.R.); bokyung12345@korea.ac.kr (B.S.); 2Advanced Food Safety Research Group, School of Food Science and Technology, Chung-Ang University, Anseong-si 17546, Korea; syjo@knacon.co.kr; 3KnA Consulting, Yongin-si 16942, Korea; shchoi@knacon.co.kr; 4Department of Statistics, Dongguk University-Seoul, Seoul 04620, Korea; yongsungjoo@dongguk.edu

**Keywords:** thallium, trace element, food contamination, risk assessment, ICP-MS

## Abstract

Thallium (Tl) is a rare element and one of the most harmful metals. This study validated an analytical method for determining Tl in foods by inductively coupled plasma mass spectrometry (ICP-MS) based on food matrices and calories. For six representative foods, the method’s correlation coefficient (R^2^) was above 0.999, and the method limit of detection (MLOD) was 0.0070–0.0498 μg kg^−1^, with accuracy ranging from 82.06% to 119.81% and precision within 10%. We investigated 304 various foods in the South Korean market, including agricultural, fishery, livestock, and processed foods. Tl above the MLOD level was detected in 148 samples and was less than 10 μg kg^−1^ in 98% of the samples. Comparing the Tl concentrations among food groups revealed that fisheries and animal products had higher Tl contents than cereals and vegetables. Tl exposure via food intake did not exceed the health guidance level.

## 1. Introduction

Since the industrial revolution, modern technological development and associated environmental pollution have increased the exposure to trace metals. With novel applications in microelectronics, chemotherapies, and other emerging technologies, the toxicological significance of some uncommon or infrequently used metals has risen [1]. In this regard, thallium (Tl) is rare but is one of the most harmful metals. William Crookes discovered it in 1861, and the metallic form was first obtained by Lamy in 1862 [2]. The name is derived from the Greek word “thallos,” which means young green twig [2]. Tl can be discovered in pure form or in combination with other elements. Pure Tl has no odor or taste. It is utilized in the production of optic lenses and glasses and the semiconductor and pharmaceutical industries [3]. Tl was used as a rodenticide and insecticide until 1972 but was prohibited because of its potential toxicity to humans [4]. Because the Tl ion charge and radius are similar to those of potassium ion, Tl can interfere with the biological functions of potassium [5]. Gastroenteritis, polyneuropathy, and alopecia are three significant symptoms of Tl poisoning in humans [6]. Alopecia is a specific symptom of Tl poisoning. Studies of Tl toxicity in humans include case studies, clinical reports, and medical investigations. Epidemiological studies associated with long-term exposure to Tl are restricted by small scales and insufficient information. The average lethal oral dose in adults has been estimated to be 10–15 mg kg^−1^ [7]. If not treated, it generally takes 10–12 days to die, but there are reports of death within 8–10 h [8].

The regulation for Tl sets a maximum contaminant level (MCL) of 0.2 μg L^−1^ in drinking water in the United States [9], and 0.8 μg L^−1^ in fresh water and 1 mg kg^−1^ dry weight in soil in Canada [10,11]. In South Korea, food regulations for most common toxic metal(loid)s such as cadmium, lead, mercury, and arsenic are established and managed to ensure safety throughout food consumption [12]. In contrast, preventive research and analytical method verification are necessary to avoid food accidents and respond to possible hazards from excess Tl.

Heavy metal analysis can be performed using such analytical techniques as inductively coupled plasma atomic emission spectrometry (ICP-AES), graphite furnace atomic absorption spectrometry (GF-AAS), and inductively coupled plasma mass spectrometry (ICP-MS). The latter is suitable for trace element analysis because of its high resolution and two to three times lower detection limit than ICP-AES [13]. In recent years, the number of studies on food analysis using ICP-MS has increased [14,15,16,17,18]. Given that there were no detailed studies on Tl content in various food products in the South Korean market, we validated a method for Tl determination by ICP-MS, monitored the commonly consumed foods in South Korea, and assessed the risk based on the results to provide foundational information.

## 2. Results and Discussion

### 2.1. Method Validation and Quality Control

We validated the external standard calibration curve for Tl. Regression of the correlation (R^2^) showed good linearity: above 0.999. Trueness was represented by certified reference material (CRM) (BCR-679, white cabbage). The measured average concentration was 3.03 ± 0.36 μg kg^−1,^ and the mean recovery was 101%. The relative standard deviation (SD) was 2.94%. Six representative foods were tested for accuracy and precision by adding standard solutions at low (0.1 μg kg^−1^), medium (0.5 μg kg^−1^), and high (1.0 μg kg^−1^) concentrations (Appendix A). The intraday accuracy ranged from 82.06% to 119.81%, interday accuracy—from 92.05% to 110.44%; the intraday precision—from 0.88% to 9.08%, interday precision—from 1.09% to 9.79%.

We observed a recovery reduction in the salted solid matrix representative food (sea salt) to less than 60% during the method validation. Salts (i.e., sodium, calcium, chloride, and potassium) cause interference in ICP-MS [19]. Higher matrix salt concentrations reduce the signal and lower the ionization potential, resulting in higher matrix effect [20]. Sample dilution can reduce the severity of the matrix effect and instrument drift [20,21]. It is recommended to keep the total dissolved solid level of samples less than 0.2% when using ICP-MS [21]. The original sample weight was 0.5 g, and the sample decomposition solution weight was 20 g. To reduce the salt interference effect, we adjusted the sample weight from 0.5 g to 0.1 g, and the decomposition solution was diluted to 80 g with distilled water. As a result, the accuracy of sea salt determination was restored from 92.05% to 101.42%. The traditional South Korean diet includes high amounts of sodium [22]. The salt interference effect can be reduced in evaluating foods with high salt content by diluting sample solutions to prevent low recovery.

The method limit of detection (MLOD) for representative foods ranged from 0.0070 to 0.0498 μg kg^−1^, and the method limit of quantification (MLOQ) ranged from 0.0222 to 0.1585 μg kg^−1^ (Table 1). We calculated the measurement uncertainty to a non-fatty matrix (rice) that spiked the Tl standard solution’s intermediate concentration (0.5 μg kg^−^^1^). The expanded uncertainty for rice was 0.51 ± 0.03 μg kg^−1^. The expanded uncertainty of the measurement is stated at an approximately 95% confidence level using coverage of *k* = 2. If the target uncertainty is not established in a regulation, an additional 20–30% tolerance can be considered to allow for variability in the uncertainty calculation process [23].

### 2.2. Concentration of Tl in Foods

#### 2.2.1. Concentration of Tl in Agricultural, Fishery, and Livestock Products

We measured the Tl concentration in agricultural, fishery, and livestock products commonly consumed in the South Korean market and summarized the results into 13 sub-groups considering the Korea Food Code (Table 2, Appendix A). Tl above the MLOD was detected in 120 out of 175 samples. The total number of cereal types tested was 32, with Tl exceeding MLOD in 16 of them. The mean value was 0.83 μg kg^−1,^ and the highest value was 5.27 μg kg^−1^ for sorghum, whereas Tl in rice was reported to be 26.8 μg kg^−1^ in a Polish study [24]. Tl contents in rice, barley, and oats were below MLOD in the current study. Tl levels in tuber crops, such as potatoes and sweet potatoes, exceeded the MLOD. The mean Tl content in tuber crops was 6.67 μg kg^−1^, and the highest of 13.21 μg kg^−1^ was detected in sweet potatoes. Tl content in potatoes, 4.04 μg kg^−1^, was similar to that in an Italian study (6.90 μg kg^−1^) and higher than that in a South African one (1.03 μg g^−1^) [25,26]. In the bean category, three types of beans—white, black, and kidney beans—were examined. In kidney beans, Tl content was below the detection limit. The average Tl content in beans was 0.49 μg kg^−1^, lower than the Italian study result (13.4 μg kg^−1^) [25].

Apple, mandarin, and pear were the three fruits examined. Tl levels in all mandarins were below MLOD, whereas Tl was detected in all pears. The mean concentration in the fruit group was 1.48 μg kg^−1^ (Table 2). In Italy, the contents in five fruits varied from 0.7 to 3.1 μg kg^−1^ [25], and we obtained similar values. Fifty vegetables were categorized as root, leafy, and other vegetables. In the case of root vegetables including carrot, white radish, garlic, onion, burdock root, and ginseng, Tl was detected in 14 out of 19 samples, with concentrations ranging from 0.02 to 7.11 μg kg^−1^, and the highest detected content was in white radish. The mean content reduced in the following order: white radish > ginseng > garlic > carrot > burdock root > onion. Tl was found in 14 out of 16 leafy vegetable samples, ranging from 0.02 to 15.93 μg kg^−1^, with most Tl detected in cabbage. The mean contents in lettuce and cabbage were relatively high in this group. A previous study discovered that the Brassicaceae family (cabbage, rape, turnip, and mustard) contains high levels of Tl [27]. Cabbage showed the highest concentration of Tl (4.13 ± 6.82 μg kg^−1^) in our category, although the SD across samples was large, indicating the differential Tl content in the soils where the samples were grown. Tl concentrations were below MLOD in zucchini and cucumber and were 0.60 and 0.49 μg kg^−1^ in one sample of red pepper and eggplant, respectively. Tl concentrations in native plants were approximately 0.05 mg kg^−1^ [28], and in Chinese crops, they ranged from 0.01 to 0.06 mg kg^−1^ [29].

In the case of livestock products, Tl in chicken, beef, and pork ranged from 0.51 to 5.72 μg kg^−1^ with the mean value of 2.46 μg kg^−1^. The highest Tl level among animal products was in eggs, with the mean value of 2.46 μg kg^−1^, more than that in a Belgian study [30]. Tl in eggs came endogenously from chickens, and the natural environment in which they grew up was conceivable to contribute [30,31]. The mean concentration in meats was 0.79 μg kg^−1^, similar to that reported in a Swiss study [32].

Fishery products were classified into fish, cephalopodan, shellfish, crustacean, and sea alga products. Tl content was higher than the detection limit in 44 out of the 50 samples tested in this category (Table 2). Tl contents in fishery products ranged from 0.02 to 12.72 μg kg^−1^, with a total average concentration of 1.55 μg kg^−1^. Tl levels in fishes varied from 0.41 to 1.87 μg kg^−1^, with the mean value of 1.06 μg kg^−1^. These values were higher than those in a Vietnamese study (0.012–0.050 μg g^−1^), but similar to those in the Canadian (0.01–19.26 ng g^−1^) and Iranian (0.02–0.30 μg kg^−1^) ones [33,34,35]. As for cephalopods, we analyzed squid and octopus, and their average Tl content was 1.10 μg kg^−1^. Crustaceans examined were four types of crabs, with an average Tl concentration of 1.28 μg kg^−1^. The average Tl content in shellfish was 2.55 μg kg^−1^, with the highest concentration (6.02 μg kg^−1^) observed in the manila clam. Sea algae were investigated in three types: laver, sea mustard, and hijiki (Appendix A). The hijiki contained the least quantity of Tl, with Tl concentrations of 5.21 and 12.72 μg kg^−1^ in one sample of laver and sea mustard samples, respectively. The SD of Tl levels among sea algae and shellfish was higher than in other species. Since each sample originated from a different origin site, such a large SD might represent habitat variations related to the marine environment. The Tl content in aquatic creatures is influenced by seawater along the coast they inhabit. The Tl concentration in seawater is about 0.2–20 μg kg^−1^, affected by volcanic events, airborne particles, and deposit inflows from rivers and soil [36]. As a result, if polluted river water or sediment reaches a habitable area, it may affect the Tl concentration of aquatic species. The presence of Tl in sediment or soil deposits enables the metal to move into benthic species such as shellfish (zoobenthos) and sea algae (phytobenthos) rather than fish and cephalopods due to their habiting nature. In recent years, numerous and broad investigations on Tl contamination have been undertaken in China [37,38,39,40,41]. There are Tl deposits in India, Japan, and Uzbekistan in Asia, but China has the most Tl deposits (i.e., Lanmuchang, Nanhua, Yunfu, and Chengmenshan deposits) [42]. The presence of Tl in crops is related to its level in the soil. Tl levels in the crust range from 0.1 to 1.7 mg kg^−1^, with sulfide ores being the most common source [43]. Lee. et al. conducted a study on soil in South Korean locations and reported the estimated Tl content of 1.20–12.91 mg kg^−1^ in the soil near a cement industry facility, and 0.18–1.09 mg kg^−1^ near a smelter and mine [44]. In addition, 2.28 ± 1.39 mg kg^−1^ of Tl was found in rhizospheric soils near cement plants in China [45].

Although the Tl content of commonly consumed foods in South Korea is low, additional studies on areas with potential Tl contamination are required, considering the concentration of Tl in contamination-prone parts of the country.

#### 2.2.2. Concentration of Tl in Processed Foods

The processed foods were monitored on 129 samples of commercial products consumed by the general population (Table 3). Canned foods were excluded because of the possibility of heavy metal cross-contamination from packaging. Tl was detected in 28 out of 129 samples of beverages and other processed foods, with considerable Appendix A.

Beverages were subdivided into infusion tea and other beverages (e.g., soda, coffee, and sports drinks). Other processed foods included various processed products, such as oils, snacks, and salted foods (e.g., sauces, pickled foods, salted seafood, and ham). In the beverage category, Tl was detected in 14 out of 39 samples. Tl concentration in tea infusions ranged from 0.01 to 1.99 μg kg^−1^. Matte tea had the highest average concentration (1.50 μg kg^−1^), followed by Solomon-seal (*Polygonatum odoratum var. pluriflorum*) and black tea (Appendix A). Our results were lower than the Tl concentrations in herbal tea infusions reported in Brazil [46]. Regarding other beverages, the average Tl concentration in red ginseng drink was 2.49 μg kg^−1^, followed by fruit juice (0.68 μg kg^−1^) and soy milk (0.24 μg kg^−1^). Samples with mainly undetected Tl and limited sample size were classified as other processed foods (Appendix A). Potato crisps and fruit jelly were categorized as snacks. The average Tl concentration in potato crisps was 3.49 μg kg^−1^, much lower than in a New Zealand study (0.17 mg kg^−1^) [47]. The average Tl level in fruit jelly was 0.51 μg kg^−1^. In the processed salty seafood category, Tl was only detected in fish cakes (1.19 μg kg^−1^). In the pickled food category, one of the three radish kimchi samples had a Tl content of 19.46 μg kg^−1^, indicating that the Tl found in the sample might have come from the raw material. Tl was not detected in sauces, oil and dairy products, cereals, livestock processed foods, and noodles.

In our study, processed foods showed a lower Tl detection frequency than raw materials. Processed foods have lower raw material contents and are supplemented with water and various food additives. Certain processed foods require the use of food extracts rather than raw foods. As a result, the metal concentration in the final product was somewhat diluted. Furthermore, cooking processes, such as blanching, boiling, and frying, can lower the amount of metal in food. Several previous studies have reported this trend [48,49]. For these reasons, heavy metal levels in processed foods are often low. Toxic metal(loid)s (including nickel, chromium, and manganese) can contaminate cooking utensils or packaging processes [50]. Contrastingly, Tl in foods can come from raw materials cultivated in a polluted environment (i.e., soil and river).

### 2.3. Comparing the Concentrations of Tl among Food Groups

We divided the food groups based on biological similarities and compared the intragroup and intergroup differences in Tl concentration (Figure 1). The classification is as follows: cereals and beans (CB), fruits and vegetables (VF), livestock products (LP), and fishery products (FP). The processed foods were excluded from the comparison because Tl was not detected in most of their samples.

When cereals and beans were compared, the median Tl value of beans was higher than that of cereals, but the difference was not statistically significant (Figure 1a). Legumes are the fruits of plants in the family Fabaceae, while cereals are fruits (caryopsis) with a seed coat (testa). The fact that there was no significant difference between the groups might be due to their comparable morphological forms. In the VF group, vegetables were subdivided into root vegetables, leafy vegetables, tuber crops, and other vegetables. The Tl concentration reduced in the following order: tuber crops > root vegetables > leafy vegetables > fruits > other vegetables (Figure 1b). A tuber is a nutrient storage organ that grows underground and belongs to the root type, but we separated it based on the classification standard of the Korea Food Code [12] and monitored potato (Solanum tuberosum L. ) and sweet potato (Ipomoea batatas) in the group (Appendix A). The Tl concentration in sweet potato (9.29 μg kg^−1^) was higher than that of potato (4.04 μg kg^−1^), given that potato is a stem tuber crop and sweet potato is a root tuber crop. In a recent Chinese study, the bio-concentration factor (BCF) and transfer factor (TF) of Tl in sweet potato were more than one, indicating that the plant can accumulate metals [51]. The median Tl concentration in root vegetables was negligibly higher than that in leafy vegetables and fruits. In the other vegetable group, which included eggplant, red pepper, zucchini, and cucumber, Tl content was significantly (*p* < 0.05) lower than in the other VF groups. Based on the biological classification, fishery products were divided into fishes, cephalopods, shellfish, crustaceans, and sea algae. Tl levels in aquatic animals were significantly (*p* < 0.05) higher than those in aquatic plants among the FP groups (Figure 1c). While there was no significant difference between the aquatic animals, the shellfish group had the highest median Tl value. Livestock products were classified as eggs and meats. The median value of eggs was insignificantly higher than that of meats (Figure 1d).

Examining the significant differences among the CB, VF, LP, and FP groups revealed that Tl content decreased in the following order: LP and FP > VF > CB (*p* < 0.05). Tl concentrations in animal foods (LP and FP) were significantly (*p* < 0.05) higher than those in the CB and VF groups (Figure 1e). Fisheries and livestock products belong to the upper trophic level of the food chain. Although sea algae are in the marine product group, Tl concentration in animals is apparently higher than that in plants. In other words, it seems that Tl accumulation can occur as it goes up the food chain. Toxic metal(loid)s accumulate in the food chain in natural environments [52,53]. Tl is enriched in the natural environment through anthropogenic sources (e.g., cement production and coal combination) and natural sources, such as volcanic activity and mineral ore mining [36]. Tl accumulates in plants and animals through the food chain, eventually causing harm to humans.

### 2.4. Health Risk Assessment for Thallium through Food Consumption

We evaluated Tl exposure through food intake based on the monitoring results. Food intake and the average weight by age were calculated from the National Health and Nutrition Survey in Korea (KNHANES) data [54]. The information on chronic exposure to Tl in the human impact case study is limited because it is mainly associated with unintentional intake or suicide attempts. A recent study by Andrew et al. used a health-based guidance value (HBGV) approach based on the tolerable daily intake (TDI) of the Netherlands National Institute for Public Health and the Environment (RIVM) while considering human toxicological uncertainty [47,55]. In the present study, the same method was applied to evaluate the dietary exposure and risk assessment for Tl.

Table 4 displays the exposure and risk assessment results based on the Tl concentrations in foods. The exposure through food consumed was very low, below 0.0032 μg kg^−1^ bw day^−1^ for all food groups. Among them, the most exposed food group was vegetables (0.0000–0.0032 μg kg^−1^ bw d^−1^), followed by tubers (0.0012–0.0029 μg kg^−1^ bw d^−1^), eggs (0.0003–0.0028 μg kg^−1^ bw d^−1^), and fruits (0.0000–0.0011 μg kg^−1^ bw d^−1^). We estimated the dietary exposure level specifically by age in both lower bound (LB) and upper bound (UB) (Appendix A). At 0–2 years, Tl exposure through vegetables was the highest, with 0.0000–0.0055 μg kg^−1^ bw d^−1^. The exposure through eggs reached a maximum level at 3–6 years (0.0010–0.0086 μg kg^−1^ bw d^−1^), 7–12 years (0.0006–0.0052 μg kg^−1^ bw d^−1^), and 13–19 years (0.0003–0.0028 μg kg^−1^ bw d^−1^), respectively. In adults aged 20–64 years, exposure through vegetables ranged from 0.0000 to 0.0031 μg kg^−1^ bw d^−1^, and in the people aged 65 years and older, tubers were the highest at 0.0009–0.0043 μg kg^−1^ bw d^−1^. Because our detection limits and Tl concentrations were low, the difference between LB and UB was minimal. Thus, we reported the exposure dose and % HBGV based on the UB (Table 4).

In all age and dietary groups, no exposure estimation surpassed the health-based guidance value. Our results indicate that the risk of exposure through Tl-contaminated food ingestion is at a minimum level.

## 3. Materials and Methods

### 3.1. Instrumentation

Tl was determined using an Agilent 7700x (Agilent Technologies, Santa Clara, CA, USA) ICP-MS with ion monitoring at mass-to-charge ratios (*m*/*z*) 203 and 205 to gather the data. The specific instrument parameters and analytical conditions are described in Appendix A.

### 3.2. Reagents and Solutions

The reagents used for this study were of analytical grade, and the water was deion-ized and purified below 18.2 MΩ using a YL WPS System (Young Lin Instrument Co., Anyang-si, Kyunggi-do, Korea). As a standard solution for Tl, the periodic table mix 1 for the ICP solution (Sigma-Aldrich, St. Louis, MO, USA) was diluted with 2.5% nitric acid to 100 μg kg^−1^ of stock solution. Analytical grade nitric acid (Chemitop, Jin-cheon, Cheongbuk-do, Korea) and hydrogen peroxide (Chemitop, Jincheon, Cheongbuk-do, Korea) were used. BCR-679 (white cabbage) certified reference materials (CRM) for method validation and quality assurance were purchased from the European Commission (European Com-mission, Joint Research Centre, Institute for Reference Materials and Measurements, Geel, Belgium).

### 3.3. Sample Collection and Processing

According to KNHANES, monitored samples were chosen to represent commonly consumed foods. Additionally, we referred to a report on the reassessment of toxic metal(loid)s criteria in South Korean food published by the Korea Ministry of Food and Drug Safety (MFDS) [54,56]. For representativeness, agricultural (113 samples), animal (12 samples), and fishery products (50 samples) were purchased from more than three distinct locations. For processed foods (129 samples), more than three products were purchased per manufacturer. Before homogenizing, the samples were washed with deionized water, and the surface moisture and non-edible components (e.g., peels, seeds, and bones) were removed. Fruits and vegetables were dried at 65 °C for 18 h before homogenization with a lab blender for 60 s and sealed at −18 °C. For infusion teas, 1.2 g of tea sachet was immersed in 100 mL of distilled water at 80 °C for 2 min. We used the infused water as the samples.

### 3.4. Sample Preparation

The current study’s sample preparation procedure was based on the elemental analysis manual of the US Food and Drug Administration (US FDA) [57] and guidelines for the analysis of toxic metal(loid)s in food by MFDS [58]. We classified foods into six categories based on their energy (i.e., calories) and matrix. The calorific value of each food was determined using the MFDS food nutrient database [59]. The classification included non-fatty solids, hydrated solids, fatty solids, salty solids, non-fatty liquids, and fatty liquids. Table 5 lists the sample weights for various sample classifications. The samples were pre-decomposed on a hot plate with 4 mL of nitric acid (70 vol%) and 1 mL of hydrogen peroxide (30 vol%) in a polytetrafluoroethylene (PTFE) vessel. After cooling at 25 °C, 3 mL of nitric acid (70 vol%) was added. PTFE vessels were sealed and placed in a microwave digestion system (ETHOS Easy, Milestone, Italy). After digestion, the samples were cooled at 25 °C and diluted to 20.0 g (in the case of salty foods, they were diluted to 80.0 g) with distilled water. All samples were analyzed in triplicates. The hydrated solid foods were converted to wet-based concentrations by considering the moisture content before drying.

### 3.5. Single-Lab Validation and Quality Control

Food comprises various organic components. In the method validation for determining toxic metal(loid)s, the matrix effect is one of the critical factors in ICP-MS analysis. We selected six representative foods for each category, taking into account the effects of matrices, such as salts and fats, and validated the method for determining Tl. We confirmed the factors for method validation in terms of linearity, trueness, accuracy, and precision. The linearity for external standard calibration curves was measured in stock solution diluted to 0.025, 0.05, 0.1, 0.25, 0.5, 1.0, and 2.0 μg kg^−1^. The trueness of Tl determination was verified using CRM Tl mass fractions. The measured values were compared with the certification values seven times. The accuracy and precision were calculated intraday and interday. The standard solution was spiked into each representative sample, resulting in the final concentrations of 0.1 μg kg^−1^ (low concentration), 0.5 μg kg^−1^ (medium concentration), and 1.0 μg kg^−1^ (high concentration) in seven replicates, respectively. The measurement uncertainty was calculated for the rice matrix spiked with a medium concentration of Tl standard solution. The sources of measurement uncertainty were expressed in the form of a fishbone diagram (Appendix A).

The MLOD and MLOQ were determined using the standard addition method, which involved adding five different Tl concentrations (0.05, 0.1, 0.25, 0.5, and 1.0 μg kg^−1^) to representative foods for calibration curve with seven repetitions. The equations are as follows:(1)MLOD=3.14× S/σ and MLOQ=10 ×S/σ
where 3.14 = student *t* factor for seven replicates, S = slope mean, and σ = standard deviation of intercept. For quality control, CRM was analyzed for every batch of samples, and recovery was checked within 80–120%. 

### 3.6. Statistical Analysis

The analysis data were processed using SPSS 21 (IBM Corporation, New York, NY, USA) to compare the levels among the food groups. Significant differences were defined using t-tests or one-way ANOVA. Statistical significance was set at *p* < 0.05. For statistical calculation, values below MLOD were replaced with half the MLOD values, and its natural logarithm was used.

### 3.7. Risk Assessment and Exposure to Thallium

Risk assessment was conducted based on monitoring data, daily consumption value, and the average body weight derived from KNHANES. To evaluate the risk of potential exposure, we used the HBGV approach. In the present study, % HBGV was calculated using 0.2 μg kg^−1^ bw day^−1^ as the HBGV of Tl [46]. Estimated daily intake (EDI) and health risk (% HBGV) were determined using the following equations:(2)$$\mathrm{EDI}\;\left(\mu g/\mathrm{kg}\;\mathrm{bw}/\mathrm{day}\right)=\frac{\mathrm{Daily}\;\mathrm{consumption}\;\left(g/\mathrm{day}\right)\times \mathrm{Tl}\;\mathrm{concentration}\;\left(\mu g/\mathrm{kg}\right)}{\mathrm{Average}\;\mathrm{body}\;\mathrm{weight}\;\left(\mathrm{kg}\right)}$$
where average body weights (kg) by age are: 0 to 2 years (11.52 kg), 3 to 6 years (18.75 kg), 7 to 12 years (35.76 kg), 13 to 19 years (59.72 kg), 20 to 64 years (64.93 kg), over 65 years (60.20 kg).
(3)$$\%\;\mathrm{HBGV}=\frac{\mathrm{EDI}}{\mathrm{HBGV}}\times 100$$

## 4. Conclusions

ICP-MS was used to validate analytical methods for six typical foods selected based on food matrices and calories. Tl levels were monitored in 304 commonly consumed food items. As a result, Tl was detected in 148 samples, with 97% of the measured samples containing less than 10 μg kg^−1^ Tl. We examined Tl levels in agricultural, fishery, and livestock products and found that Tl concentrations in animal foods are higher than in plant foods. The results confirmed that the level of Tl exposure through food intake was within the health-based guidance value. Further research and regular monitoring of Tl in foods are necessary to protect human health and avoid food-borne contamination.

## Figures and Tables

**Figure 1 molecules-26-06729-f001:**
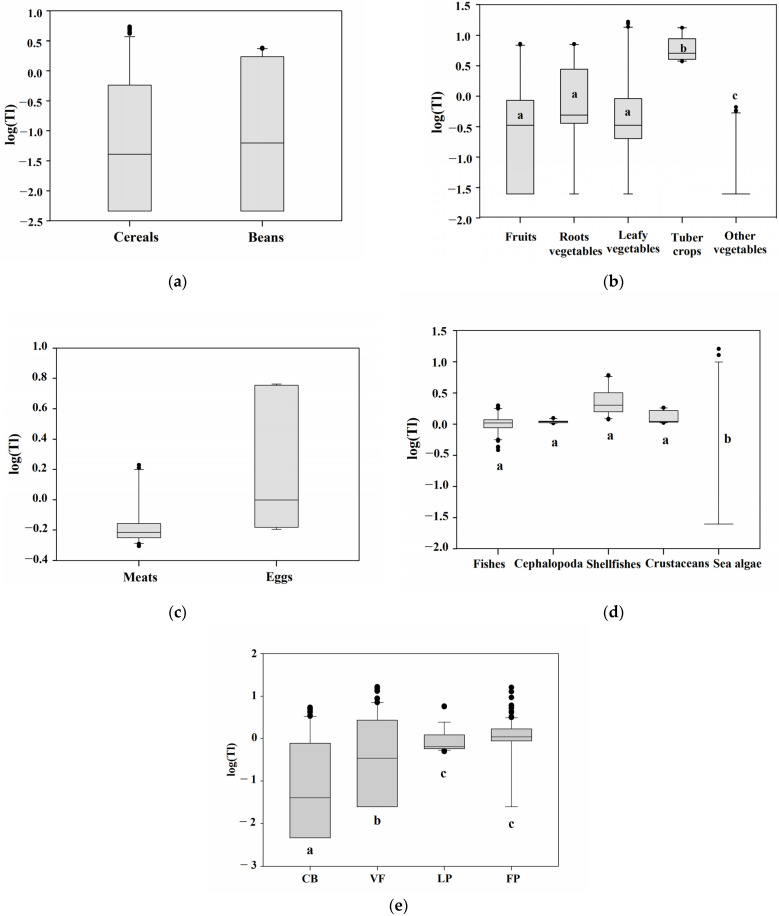
Box plots of thallium (Tl) in food groups by categories. (**a**) cereals and beans, (**b**) fruits, vegetables, and tubers, (**c**) livestock products, (**d**) fishery products, (**e**) CB, cereals and beans; VF, vegetable and fruits; LP, livestock products; FP, fishery products (line = median, box = 25th and 75th percentiles, whiskers = minimum and maximum observations below upper fence); IQR (Interquartile Range) 75th–25th percentile. Means within box plots with different letters are significantly different at *p* < 0.05 by testing between mean using *t*-test or one-way ANOVA with Tukey’s multiple comparison.

**Table 1 molecules-26-06729-t001:** Selection of representative foods and validation parameters of Tl analysis.

Classification	Representative Food	Regression Equation	R^2^	MLOD ^a^ (μg kg^−1^)	MLOQ ^b^ (μg kg^−1^)
Non-fatty Solid	Rice	y = 117366x − 208.52	1	0.0092	0.0293
Hydrated Solid	Apple	y = 38252x + 774.86	0.9999	0.0498	0.1585
Fatty Solid	Beef	y = 85346x − 6.9005	0.9997	0.0070	0.0222
Salty Solid	Sea Salt	y = 67663x + 165.49	0.9999	0.0207	0.0661
Non-Fatty Liquid	Orange Juice	y = 88399x − 441.29	1	0.0126	0.0401
Fatty Liquid	Sesame Oil	y = 111833x − 120.50	1	0.0225	0.0715

^a^ MLOD refers to the method limit of detection, S/σ = 3.14; ^b^ MLOQ refers to the method limit of quantification, S/σ = 10.

**Table 2 molecules-26-06729-t002:** Concentration (fresh weight) and detection frequency of Tl in agricultural, fishery, and livestock products.

Category	Subcategory (n) ^a^	Frequency of Detection ^b^ (%)	Min–Max ^c^ (μg kg^−1^)	Median (μg kg^−1^)	Mean ± SD ^c^ (μg kg^−1^)
Agricultural Products	Cereals (32)	50	0.00–5.27	0.20	0.83 ± 1.47
	Beans (10)	50	0.00–2.31	0.00	0.49 ± 0.80
	Tuber crops (6)	100	3.78–13.21	5.13	6.67 ± 3.70
	Vegetables (50)	66	0.02–15.93	0.34	1.59 ± 3.21
	Mushrooms (6)	0	0.02–0.02	0.02	0.02 ± 0.00
	Fruits (9)	56	0.02–7.00	0.34	1.48 ± 2.39
Livestock Products	Eggs (3)	100	0.65–5.72	1.03	2.46 ± 2.31
	Meats (9)	100	0.51–1.63	1.00	0.79 ± 0.38
Fishery Products	Fishes (21)	100	0.41–1.87	1.03	1.06 ± 0.37
	Cephalopoda (6)	100	1.04–1.22	1.07	1.10 ± 0.06
	Shellfishes (10)	100	1.20–6.02	1.88	2.55 ± 1.46
	Crustaceans (4)	100	1.07–1.79	1.14	1.28 ± 0.29
	Sea Algae (9)	22	0.02–12.72	0.02	1.21 ± 1.40

^a^ n is the number of samples; ^b^ frequency of detection is the percentage of samples above the method detection of limit (MLOD) in each category; ^c^ results < MLOD were set equal to 1/2 MLOD values.

**Table 3 molecules-26-06729-t003:** Concentration (fresh weight) and detection frequency of Tl in processed foods.

Category	Subcategory (n) ^a^	Frequency of Detection ^b^ (%)	Min–Max ^c^ (μg kg ^−1^)	Median (μg kg ^−1^)	Mean ± SD ^c^ (μg kg ^−1^)
Beverages	Infusion Teas (18)	39	0.01–1.99	0.52	0.61 ± 0.63
	Other Beverages (21)	33	0.01–4.12	0.01	0.49 ± 0.97
Other Processed Foods	Snacks (6)	100	0.01–0.01	1.10	0.01 ± 0.00
	Salted Seafood processing Products (12)	25	0.01–1.41	0.01	0.30 ± 0.52
	Pickled Foods (12)	8	0.01–19.46	0.01	1.63 ± 5.38
	Sauces (18) Oil Products (27)	0	0.01–0.01	0.01	0.01 ± 0.00
		0	0.01–0.01	0.01	0.01 ± 0.00
	Dairy Products (6)	0	0.01–0.01	0.01	0.01 ± 0.00
	Cereal Products (3)	0	0.00–0.00	0.00	0.00 ± 0.00
	Livestock Processing Products (3)	0	0.01–0.01	0.01	0.01 ± 0.00
	Noodles (3)	0	0.01–0.01	0.01	0.01 ± 0.00

^a^ n is the number of samples; ^b^ frequency of detection is the percentage of samples above the method detection of limit (MLOD) in each category; ^c^ results < MLOD were set equal to 1/2 MLOD values.

**Table 4 molecules-26-06729-t004:** Exposure assessment and risk characterization for Tl.

Food Group		Concentration (μg kg^−1^)	Exposure Dose (μg kg^−1^ bw day^−1^)	% HBGV
Agricultural Products	Cereals	0.01–5.27	0.0000–0.0001	0.0000–0.0345
	Fruits	0.01–7.00	0.0000–0.0011	0.0000–0.5625
	Beans	0.01–2.32	0.0000–0.0001	0.0000–0.0549
	Mushrooms	0.01	0.0000–0.0000	0.0001–0.0002
	Tuber Crops	3.78–13.21	0.0012–0.0029	0.6050–1.4682
	Vegetables	0.01–15.93	0.0000–0.0032	0.0001–1.5807
Livestock Products	Eggs	0.65–5.72	0.0003–0.0028	0.1604-1.4109
	Meats	0.51–1.63	0.0001–0.0009	0.0502-0.4326
Fishery Products	Crustaceans	1.07–1.79	0.0000–0.0000	0.0130-0.0216
	Fishes	0.41–1.87	0.0000–0.0001	0.0027-0.0440
	Cephalopoda	1.04–1.22	0.0000–0.0001	0.0143-0.0445
	Shellfishes	1.20–6.02	0.0000–0.0001	0.0106-0.0606
	Seaweeds	0.01–12.72	0.0000–0.0002	0.0000-0.0900
Processed Products	Beverages	0.01–4.12	0.0000–0.0003	0.0001-0.1549
	Other Processes Products	0.00–19.46	0.0000–0.0014	0.0000-0.6881

HBGV is the health-based guidance value: % HBGV = (estimated daily intake/HBGV) × 100% HBGV was calculated based on HBGV of Tl = 0.2 μg kg^−1^ bw day^−1^.

**Table 5 molecules-26-06729-t005:** Classification into food matrix groups and sample weights for preparation.

Classification	Sample Weight (g)	Food Group
Non-Fatty Solid	0.5	Cereals, Tubers, Beans, Snacks, Cereal Products
Hydrated Solid	0.5	Fruits, Vegetables, Mushrooms, Sea Algae
Fatty Solid	0.5	Meats, Eggs, Fishes, Cephalopods, Shellfishes, Crustaceans
Salty Solid	0.1	Sauces, Pickled Foods, Salt-Added Products, Noodles
Non-Fatty Liquid	0.5	Infusion Teas, Other Beverages
Fatty Liquid	0.3	Oil Products, Dairy Products

## Data Availability

Not applicable.

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
