# Peer review of "Method Validation for Determination of Thallium by Inductively Coupled Plasma Mass Spectrometry and Monitoring of Various Foods in South Korea"

_molecules, 2021, doi:10.3390/molecules26216729_

Round 1

Reviewer 1 Report

The topic is interesting and the number of analysed samples is hight. However the article could be improved. Please see my attached comments

Author Response

Reviewer 1

Comments about entire article:

The topic is interesting, and the number of analysed samples is hight. However the article could be improved. Please see my attached comments

à First, we would like to express our gratitude to the reviewers for encouraging us to make a positive move forward. With highlighted in green color text, we made the correction in response.

- There are a lot of unclear sentences: the following are only some examples, please review all article.

à We reviewed all article with our best effort and revised thoroughly. The English grammar in this manuscript has been thoroughly reviewed and corrected.

- Line 36: “The name…. “ perhaps a verb is missing

à We made the correction in text such as in line (L) 34-35 as follows:

“The name is derived from the Greek word "thallos," which meaning young green twig [2].”

- Lines 59-61: the meaning of the sentence is unclear

à We made the correction in text such as in L 58-59 as follows:

“The latter is suitable for trace element analysis because of its high resolution and two to three times lower detection limit than ICP-AES [13].”

- Lines 121-122: perhaps a subject is missing

à We made the correction in text such as in L 120-121 as follows:

“Tl levels in all mandarins were below MLOD, whereas Tl was detected in all pears.”

- Line 131: please review the sentence

à We made the correction in text such as in L 128-129 as follows:

“Tl was found in 14 out of 16 leafy vegetable samples, ranging from 0.02 to 15.93 mg kg -1, with most Tl detected in cabbage.”

- Line 241: perhaps is decrease is not the correct form

à We made the correction in text such as in L 237-240 as follows:

“Legumes are the fruits of plants in the family Fabaceae, while cereals are fruits (caryopsis) with a seed coat (testa). The fact that there was no significant difference between the groups might be due to their comparable morphological forms.”

- Line 275: the sentence is unclear

à We made the correction in text such as in L 271-272 as follows:

“Tl accumulates in plants and animals through the food chain, eventually causing harm to humans.”

- Lines 325-328: the sentences are unclear

à We made the correction in text such as in L 319-324 as follows:

“Analytical grade nitric acid (Chemitop, Cheongbuk-do, Republic of Korea) and hydrogen peroxide (Chemitop, Cheongbuk-do, Republic of Korea) were used. BCR-679 (white cabbage) certified reference materials (CRM) for method validation and quality assurance were purchased from the European Commission (European Commission, Joint Research Centre, Institute for Reference Materials and Measurements, Geel, Belgium).”

- Please use the third person to explain your experiments

à An English native speaker at Editage Company checked this work for correct English use (www. Editage.com).

2: results and discussion

2.2.1

- Lines 87-88: the sentence is unclear. Please review it and write clearer as is written in line 355

à We appreciate the reviewer’s comment. We revised as commented in L 84-86 as follows:

“To reduce the salt interference effect, we adjusted the sample weight from 0.5 g to 0.1 g, and the decomposition solution was diluted to 80 g with distilled water.”

- Line 96: I’m sorry but I didn’t understand why you set an uncertainty with an uncertainty (0.51±0.03µg/kg) perhaps it is the difference among three concentrations used to validate? What uncertainty shall you apply?

à We made the correction in text such as in L 92-94 as follows:

“We calculated the measurement uncertainty to a non-fatty matrix (rice) that spiked the Tl standard solution’s intermediate concentration (0.5 μg kg -1). The expanded uncertainty for rice was 0.51 0.03 μg kg -1.”

2.2.2

- Lines 198-201: it is not clear if the Tl concentration was evaluated in herbs before the infusion or in the final product (tea). If the second hypothesis is correct, please in the material and method section explain how you obtained teas (the weight of used herbs, the water temperature, the infusion time, and so on): in practice the dilution factor

à We made the correction in text such as in L 336-338 as follows:

“For infusion teas, 1.2 g of tea sachet was immersed in 100 mL of distilled water at 80°C for 2 min. We used the infused water as the samples.”

- Lines 223-225: Perhaps the reduction of metal contents is caused by the dilution of raw material. I think that the processing doesn’t reduce the metal content. As you explain later for analysis you used raw materials “ready to use” (washed, cutted and so on). So please explain better this topic.

à We made the correction in text such as in L 219-225 as follows:

“In our study, processed foods showed a lower Tl detection frequency than the raw materials. Processed foods have lower raw material contents and are supplemented with water and various food additives. Certain processed foods require the use of food extracts rather than raw foods. As a result, the metal concentration in the final product was somewhat diluted. Furthermore, cooking processes, such as blanching, boiling, and frying, can lower the amount of metal in food. Several previous studies have reported this trend [48,49]. For these reasons, heavy metal levels in processed foods are often low.”

- Lines 228-231: This problem could be present for other elements such as Ni, or Cr or Mn, not for Tl…

please explain better.

à We made the correction in text such as in L 226-228 as follows:

“Toxic metal(loid)s (including nickel, chromium, and manganese) can contaminate cooking utensils or packaging processes [50]. Contrastingly, Tl in foods can come from raw materials cultivated in a polluted environment (i.e., soil and river).”

- Lines 240-241: Rice is an example for distribution I suppose… So try to explain this in your work

other wise the sentence in not useful

à We made the correction in text such as in L 237-240 as follows:

“Legumes are the fruits of plants in the family Fabaceae, while cereals are fruits (caryopsis) with a seed coat (testa). The fact that there was no significant difference between the groups might be due to their comparable morphological forms.”

- Table4: since the differences between lowerbound and upperbound are insignificant I suggest to explain better this in the work and simplify the table with only the upperbound results.

à Please see the revised Table 4. And we revised the relevant sentences in L 296-300 as follows:

“Because our detection limits and Tl concentrations were low, the difference between LB and UB was minimal. Thus, we reported the exposure dose and % HBGV based on the UB (Table 4).

In all age and dietary groups, no exposure estimation surpassed the health-based guidance value.”

We really appreciate the critical and useful input from the Academic Editor which were guidance for revising our manuscript.

Reviewer 2 Report

The paper entitled "Method validation for determination of thallium by Inductively Coupled Plasma-Mass Spectrometry and monitoring of 
various foods in South Korea" by Kim et al. is generally well written, which examined Tl levels in agricultural, fisheries, and livestock products based on validated measurement method, and found that Tl concentration in animal foods are greater than those in plant foods. The illustration is supported with abundant data, which is important for the thallium study. The paper is well within the scope of molecules. I suggest minor revision.

Minor concerns:

(1)Some writing errors should be corrected, for example,there should be no blank between numbers and "%"; other grammatical errors like "on" in Line 21, "were ranged" in Line 23, " was not exceeded" in Line 25, "for" in Line 64, "was ranged" in Lines 77-78, etc. please carefully double check.

(2)Line 118, " which" is suggested to be revised such as " in which".
(3)Line 52-53, "heavy metals" is suggested to be revised such as "toxic metal(loid)s".
(4)Line 59-60, two "because" in one sentence.
(5)Line 20, revise "accuracy was ranged from 82.06 % to 119.81 %".
(6)Line 62-63, the up to date references of "Environment International 146 (2021) 106207" and "Journal of Hazardous Materials 407 (2021) 
124402" with ICP-MS measurement of thallium are suggested to be included.
(7)Line 182-183, It would be interesting to compare the pertinent study on the same topic which was published on "Science of The Total 
Environment. 2021, 782:146603."
(8) In reference 7, something wrong is observed.

Author Response

Reviewer 2

Comments and Suggestions for Authors

The paper entitled "Method validation for determination of thallium by Inductively Coupled Plasma-Mass Spectrometry and monitoring of 
various foods in South Korea" by Kim et al. is generally well written, which examined Tl levels in agricultural, fisheries, and livestock products based on validated measurement method, and found that Tl concentration in animal foods are greater than those in plant foods. The illustration is supported with abundant data, which is important for the thallium study. The paper is well within the scope of molecules. I suggest minor revision.

à Answer:

First, we would like to express our gratitude to the reviewers for encouraging us to make a positive move forward. With highlighted in green color text, we made the correction in response.

Minor concerns:

(1)Some writing errors should be corrected, for example, there should be no blank between numbers and "%"; other grammatical errors like "on" in Line 21, "were ranged" in Line 23, " was not exceeded" in Line 25, "for" in Line 64, "was ranged" in Lines 77-78, etc. please carefully double check.

à We reviewed all article with our best effort and revised thoroughly. The English grammar in this manuscript has been thoroughly reviewed and corrected.

(2)Line 118, " which" is suggested to be revised such as " in which".

à We made the correction in text such as in line (L) 116-117 as follows:

“In the bean category, three types of beans – white, black, and kidney beans – were examined. In kidney beans, Tl content was below the detection limit.”

(3)Line 52-53, "heavy metals" is suggested to be revised such as "toxic metal(loid)s".

à We made the correction in text such as in L 50-52 as follows:

“In South Korea, food regulations for most common toxic metal(loid)s such as cadmium, lead, mercury, and arsenic are established and managed to ensure safety throughout food consumption [12].”

(4)Line 59-60, two "because" in one sentence.

à We made the correction in text such as in L 58-59 as follows:

“The latter is suitable for trace element analysis because of its high resolution and two to three times lower detection limit than ICP-AES [13].”

(5)Line 20, revise "accuracy was ranged from 82.06% to 119.81%".

à We made the correction in text such as in L 18-20 as follows:

“For six representative foods, the method’s correlation coefficient (R2) was above 0.999, and the method limit of detection (MLOD) was 0.0070–0.0498 μg kg -1, with accuracy ranging from 82.06% to 119.81% and precision within 10%.”

(6)Line 62-63, the up to date references of "Environment International 146 (2021) 106207" and "Journal of Hazardous Materials 407 (2021) 
124402" with ICP-MS measurement of thallium are suggested to be included.

à The up-to-date references were added.

  1. Wang, J.; Wang, L.; Wang, Y.; Tsang, D.C.; Yang, X.; Beiyuan, J.; Yin, M.; Xiao, T.; Jiang, Y.; Lin, W.; et al. 2021. Emerging risks of toxic metal(loid)s in soil-vegetables influenced by steel-making activities and isotopic source apportionment. Environment international 2021, 146, p.106207, https://doi.org/10.1016/j.envint.2020.106207.

(7)Line 182-183, It would be interesting to compare the pertinent study on the same topic which was published on "Science of The Total  Environment. 2021, 782:146603."

à We made the correction in text such as in L 177-180 as follows:

“Lee. et al. conducted a study on soil in South Korean locations and reported the estimated Tl content of 1.20–12.91 mg kg -1 in the soil near a cement industry facility, and 0.18–1.09 mg kg -1 near a smelter and mine [44]. Also, 2.28±1.39 mg kg -1 of Tl was found in rhizospheric soils near cement plants in China [45].”

  1. Zhou, Y.; Wang, J.; Wei, X.; Ren, S.; Yang, X.; Beiyuan, J.; Wei, L.; Liu, J.; She, J.; Zhang, W.; et al. 2021. Escalating health risk of thallium and arsenic from farmland contamination fueled by cement-making activities: A hidden but significant source. Sci Total Environ, 782, 146603, doi:https://doi.org/10.1016/j.scitotenv.2021.146603.

(8) In reference 7, something wrong is observed.

à Answer:

We made the correction as follows:

  1. Ferguson, T.J. Chapter 148. Thallium. In Poisoning & Drug Overdose, 6e, Olson, K.R., Ed.; The McGraw-Hill Companies: New York, NY, 2012.

We really appreciate the critical and useful input from the Academic Editor which were guidance for revising our manuscript.

Reviewer 3 Report

Dear Authors,

Your manuscript was very adequate written, presenting not only an analytical validation process, but its application.

Author Response

Reviewer 3

Comments and Suggestions for Authors

Dear Authors,

Your manuscript was very adequate written, presenting not only an analytical validation process, but its application.

à We truly appreciate your positive comments on our work.

Round 2

Reviewer 1 Report

I think the manuscript has been sufficiently improved. Now i agree with the  publication in Molecules.
Thank you very much.